

# Clinical associations and genetic interactions of oncogenic BRAF alleles

Sebastian A. Wagner[1,2,3]

[1] Department of Medicine, Hematology/Oncology, Goethe University, Frankfurt, Germany
[2] Frankfurt Cancer Institute (FCI), Frankfurt, Germany
[3] German Cancer Consortium (DKTK) and German Cancer Research Center (DKFZ), Heidelberg, Germany

## ABSTRACT

BRAF is a serine/threonine-specific protein kinase that regulates the MAPK/ERK signaling pathway, and mutations in the BRAF gene are considered oncogenic drivers in diverse types of cancer. Based on the signaling mechanism, oncogenic BRAF mutations can be assigned to three different classes: class 1 mutations constitutively activate the kinase domain and lead to RAS-independent signaling, class 2 mutations induce artificial dimerization of BRAF and RAS-independent signaling and class 3 mutations display reduced or abolished kinase function and require upstream signals. Despite the importance of BRAF mutations in cancer, the clinical associations, genetic interactions and therapeutic implications of non-V600 BRAF mutations have not been explored comprehensively yet. In this study, the author analyzed publically available data from the AACR Project GENIE to further understand clinical associations and genetic interactions of oncogenic BRAF mutations. The analyses identified 93 recurrent BRAF mutations, out of which 50 could be assigned to a functional class based on literature review. The author could show that the frequency of BRAF mutations varies across cancer types and subtypes, and that the BRAF mutation classes are unequally distributed across cancer types and subtypes. Using permutation testing-based co-occurrence analyses, the author defined the genetic interactions of BRAF mutations in multiple cancer types and revealed unexplored genetic interactions that might define clinically relevant subgroups. With non-small cell lung cancer as example, the author further showed that the genetic interactions are BRAF mutation class-specific. The presented analyses explore the properties of oncogenic BRAF mutations and will help to further delineate the complex role of BRAF in cancer.

# INTRODUCTION

The BRAF gene encodes a serine/threonine-specific protein kinase that regulates the MAPK/ERK signaling pathway and affects cell division, differentiation, and secretion. The BRAF protein is composed of three conserved regions (CR1-3) that are characteristic for the Raf family protein kinases: CR1 contains the Ras-GTP-binding domain, CR2 is a flexible serine-rich linker and CR3 contains the catalytic protein kinase domain (*Daum et al., 1994*). Binding of Ras-GTP to B-Raf induces a conformational change that leads to

Corresponding author
Sebastian A. Wagner,
swagner@med.uni-frankfurt.de

autophosphorylation and activation of the kinase domain (*Cutler et al., 1998*; *Cook & Cook, 2021*).

Mutations in BRAF gene are found in diverse types of cancer including melanoma, colorectal adenocarcinoma, non-small-cell lung cancer, papillary thyroid carcinoma and hairy cell leukemia and are considered oncogenic drivers (*Davies et al., 2002*; *Paik et al., 2011*; *Cardarella et al., 2013*; *Holderfield et al., 2014*). In hairy cell leukemia the BRAF Val600Glu (V600E) mutation is found in nearly all cases at diagnosis and considered as the causal genetic event (*Tiacci et al., 2011*). In other types of cancer, the frequency of BRAF mutations ranges from 80% in malignant melanoma to 1–5% in lung adenocarcinoma and colorectal cancer.

Based on the molecular mechanism, a classification of BRAF mutations identified in cancer samples has been proposed (*Dankner et al., 2018*): class 1 mutations mimic the phosphorylation of the activation loop and thereby lead to aberrant activation of the kinase domain. BRAF molecules with class 1 mutations can signal as monomers and independent of upstream RAS. The most frequently observed class 1 mutation is V600E, but also other mutations at the same amino acid position namely V600M, V600R, V600K and V600D are considered class 1 mutations. Class 2 mutations lead to artificial dimerization of BRAF and activation of the kinase domain. Similar to class 1 mutations, class 2 mutations signal independent of RAS, however, they activate BRAF significantly weaker than class 1 mutations. The most frequent class 2 mutations are located at amino acid position 469, 597 and 601. Also, BRAF fusions that have been identified in different types of cancer at low frequencies promote dimerization and signal similar to BRAF with class 2 point mutations (*Jones et al., 2008*; *Ross et al., 2016*). In contrast to class 1 and 2 mutations, class 3 mutations have low or absent kinase activity and require upstream RAS signaling to realize their oncogenic potential (*Wan et al., 2004*; *Heidorn et al., 2010*; *Yao et al., 2015*).

The high prevalence of BRAF mutations in cancer has spurred interest in the development of specific BRAF inhibitors: Vemurafenib was the first BRAF inhibitor approved for the treatment of metastasized melanoma harboring BRAF V600E mutations in 2011 (*Bollag et al., 2010*; *Chapman et al., 2011*). However, single agent treatment of melanoma patients with vemurafenib led to rapid development of resistance. Treatment strategies combining BRAF and MEK inhibitors can delay the development of resistance. In the following years additional BRAF inhibitors including dabrafenib and encorafenib have been developed and introduced into the clinic.

At present, BRAF inhibitors have been approved for the treatment of various types of cancer harboring BRAF class 1 mutations: For patients with advanced melanoma the BRAF and MEK inhibitor combinations dabrafenib/trametinib (*Flaherty et al., 2012*; *Robert et al., 2015*), encorafenib/binimetinib (*Dummer et al., 2018*) and vemurafenib/cobimetinib (*Ascierto et al., 2016*) can be employed. More recently it was demonstrated that the addition of the immune checkpoint inhibitor atezolizumab to vemurafenib and cobimetinib can improve the progression free survival of melanoma patients with BRAF V600E mutations (*Sullivan et al., 2019*; *Gutzmer et al., 2020*).

In colorectal cancer the combination of the BRAF inhibitor encorafenib with the EGFR monoclonal antibody cetuximab has shown promising results and was approved for the

treatment of patients with BRAF V600E mutations (*Tabernero et al., 2021*).
The combination of dabrafenib and trametinib has also been approved for the treatment of non-small cell lung cancer with BRAF V600 mutations (*Planchard et al., 2016*, *2017*). In addition, the BRAF inhibitor vemurafenib has been approved for the treatment of patients with Erdheim-Chester disease with BRAF V600 mutations and the combination of dabrafenib and trametinib is approved for treatment of anaplastic thyroid cancer with BRAF V600E mutations.

To date, there has been no regulatory approval for a targeted therapy in patients with non-V600 mutations. Different clinical trials have tried to explore treatment strategies for patients with non-V600 BRAF mutations with varying degrees of success (*Kotani et al., 2020*). In the case of BRAF class 2 mutations, multiple smaller clinical trials and case reports suggest that MEK inhibition might be an active treatment strategy (*Dagogo-Jack, 2020*).

Clinical evidence for patients with inactivating class 3 BRAF mutations is still largely missing: It has been suggested that tumors with class 3 BRAF mutations are sensitive to the inhibition of activated RAS (*Yao et al., 2017*). There is also preclinical evidence demonstrating activity of pan-RAF and CRAF inhibitors in tumors with class 2 and inactivating class 3 mutations (*Smalley et al., 2009*; *Kordes et al., 2016*; *Hoefflin et al., 2018*). Currently, multiple clinical trials are testing the efficacy of RAF inhibitors alone or in combination with MEK inhibitors in patients with inactivating BRAF mutations.

Recent large scale tumor sequencing efforts have greatly expanded the knowledge about genetic alterations in cancer. The AACR Project GENIE is an international data-sharing effort for clinical-grade, high-throughput sequencing (NGS) data. The data is collected at 18 cancer centers in the United States and Europe (*Sweeney et al., 2017*). In this study, the author surveyed oncogenic BRAF mutations across cancer types using data from the AACR Project GENIE. The presented analyses highlight that different oncogenic BRAF mutations are associated with distinct clinical features and genetic interactions.

# MATERIALS AND METHODS

## Data

For the analyses of clinical associations and genetic interactions of oncogenic BRAF mutations the AACR Project GENIE 11.0 public data set containing gene panel sequencing data from over 136,000 cancer samples from over 121,000 patients was downloaded from Sage Bionetworks Synapse (Synapse ID: syn26706564, DOI: https://doi.org/10.7303/syn26706564).

## Statistical analysis

All analyses were performed using the R software environment for statistical computing and graphics. Visualizations were created using the ggplot2 and ggpubr packages for the R software environment. All source code and data are available as supplemental data (Data S1).

### Identification of reoccurring BRAF mutations and classification of BRAF mutations

BRAF mutations were defined as reoccurring if they were found in ≥5 samples in the AACR Project GENIE dataset. The author attempted to assign all reoccurring BRAF mutations to their respective BRAF mutation class based on an extensive literature review (*Yao et al., 2017*; *Dankner et al., 2018*; *Schirripa et al., 2019*; *Lokhandwala et al., 2019*; *Lin et al., 2019*; *Johnson et al., 2019*; *Yaeger et al., 2019*; *Lei et al., 2020*; *Owsley et al., 2021*; *Sahin & Klostergaard, 2021*). Reoccurring BRAF mutations not previously described in the reviewed literature where marked as unknown significance (?).

### Frequency of BRAF mutations across cancer types

For calculation of BRAF mutation and BRAF mutation class frequencies across cancer types only samples with reoccurring BRAF mutations were counted. When multiple BRAF mutations were identified in one sample only the mutation with the highest clinical significance was considered. The clinical significance was defined as class 1 > class 2 > class 3 > unknown significance.

### Co-mutation analyses

For co-mutation analyses a permutation test was employed: initially samples were filtered according to cancer type and BRAF mutation class and samples with reoccurring BRAF mutations were labeled. For each investigated subset the absolute and relative frequencies of co-mutation of BRAF with other genes were calculated (observed absolute and relative frequency). Next the sample labels were randomly permutated and co-mutation frequencies were calculated. To generate a robust null hypothesis, random permutation was performed one million times for each investigated subset. Left and right *p* values were calculated for each gene by counting the permutations with co-mutation frequency lower or higher than the observed absolute frequency.

In the figures observed relative frequency and expected relative frequency were visualized as scatter plot. Only genes with an absolute difference between observed and expected relative frequency ≥2% and a p value corrected for multiple hypothesis testing ≤0.001 are shown.

## RESULTS

### Overview of oncogenic BRAF alleles

Recent large scale tumor sequencing efforts have greatly expanded the knowledge about genetic alterations in cancer. To investigate clinical associations and genetic interactions of oncogenic BRAF alleles, the AACR Project GENIE 11.0 public dataset containing gene panel sequencing data from over 136,000 cancer samples from over 121,000 patients was downloaded.

Initially, a list of all BRAF alleles in the dataset was compiled. On the protein level, 914 unique BRAF mutations were identified. The BRAF V600E mutation was the most frequently occurring mutation which was detected in 3,905 cancer samples. A total of 93

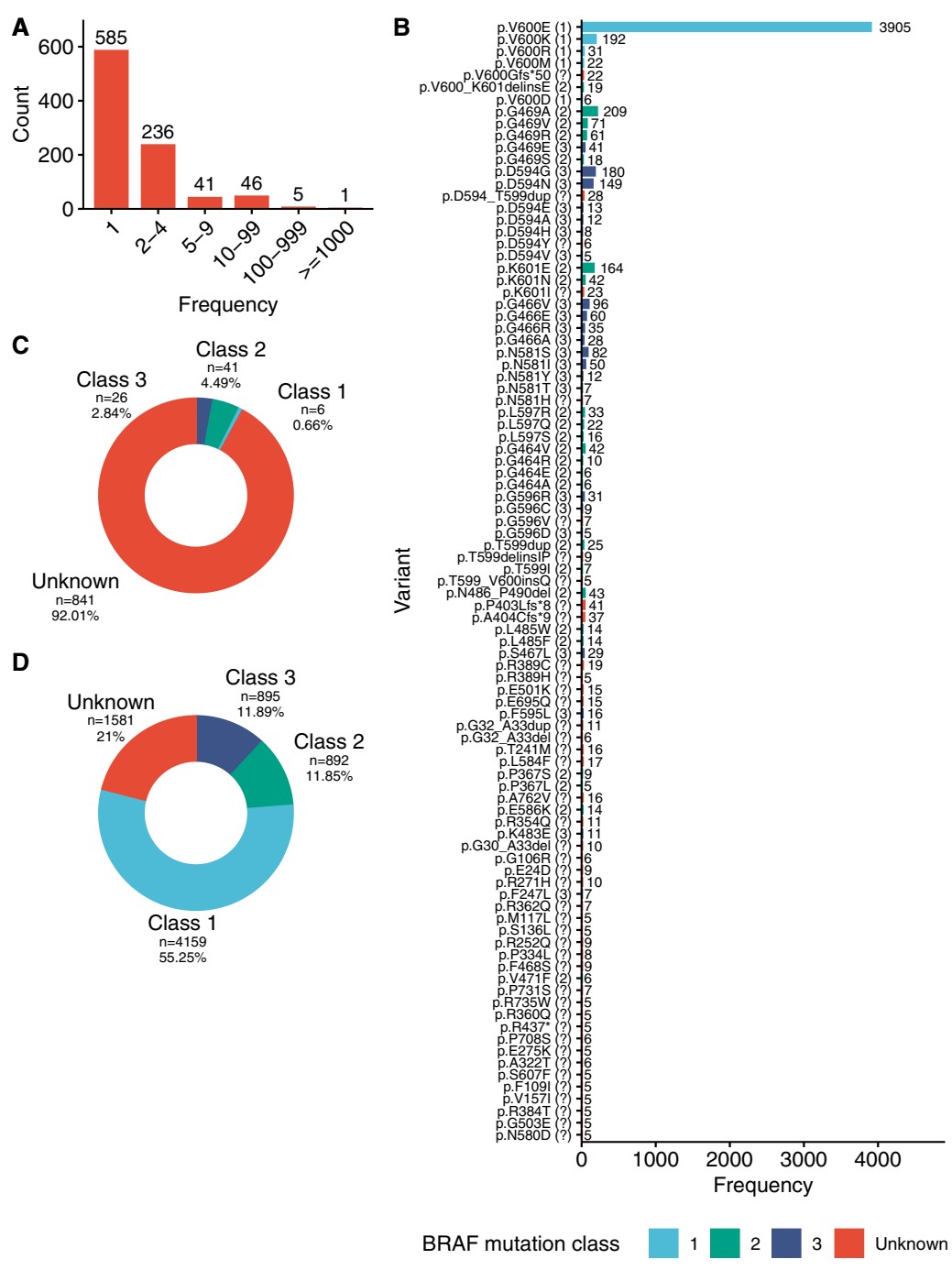

**Figure 1 Reoccurring BRAF mutations.** (A) Sample count for all identified BRAF mutations in the AACR Project GENIE dataset. (B) Frequency and class of all reoccurring BRAF mutations. (C) Count of distinct mutations assigned to the indicated mutation classes. (D) Count of samples with BRAF mutations assigned to the indicated mutation classes.

BRAF mutations could be identified in ≥5 samples and were considered reoccurring mutations (Figs. 1A and 1B). A total of 821 mutations were detected in less than five samples and out of these 585 mutations could be identified in only one sample (Fig. 1A).

Next the author conducted an extensive literature review to assign the identified BRAF mutations to their respective functional classes: six mutations (0.66%) at the amino acid position 600 were considered class 1 mutations. These mutations included V600E as well as V600D, V600M, V600R and V600K. A total of 41 mutations (4.49%) were considered class 2 mutations and included among others mutations at amino acid position 469, 601, 597 and 464. 26 mutations (2.84%) were considered class 3 mutations. Mutations at amino acid position 594 and 466 were the most frequently occurring class 3 mutations in the AACR Project Genie dataset. In total, 50 (53.76%) of 93 reoccurring mutations could be assigned to a functional class based on the literature review. A large fraction (841, 92.01%) of the BRAF variants in the dataset has not been studied in detail and could not be assigned to a specific class (Fig. 1C). However, these variants of unknown significance were observed at a lower frequency and were only present in 21% (1,581) of the samples with BRAF mutations (Fig. 1D).

## Relative frequency of BRAF mutations and BRAF mutation classes across cancer types

Next, the author investigated relative frequency of BRAF mutations in the different cancer types. For these analyses only reoccurring BRAF mutations identified in ≥5 samples were considered. The relative frequency of BRAF mutations varied greatly across the different cancer types (Fig. 2A): in thyroid cancer 39.53% (742/1,877) of the samples contained a reoccurring BRAF mutation, in melanoma 32.91% (1,809/5,496) of the samples contained a BRAF mutation, in colorectal cancer 10.38% (1,337/12,880) contained a BRAF mutation and in NSCLC 4.4% (850/19,319) contained a BRAF mutation. In other cancer types such as pancreatic cancer, prostate cancer and breast cancer, BRAF mutations were only found at low relative frequencies (1.62%, 1.54% and 0.63%, respectively). The relative frequency of BRAF mutations also varies among different cancer subtypes (Fig. 2B).

Next, the author asked if BRAF mutation classes are equally represented in different cancer types. To this end, we plotted the relative frequency of the different BRAF mutation classes for all cancer types in the project GENIE dataset (Fig. 3). These data clearly demonstrated that BRAF mutation classes are not equally distributed across cancer types but show a distinct cancer-type specific distribution. In thyroid cancer, almost all identified BRAF mutations could be assigned to class 1. Also, in melanoma and colorectal cancer, the majority of BRAF mutations could be assigned to class 1. In non-small cell lung cancer, class 1, 2 and 3 BRAF mutations were found in a similar frequency. Notably, in prostate cancer, a majority of the BRAF mutations could be assigned to class 2. In small cell lung cancer and cervical cancer no BRAF class 1 mutation could be identified, but the overall BRAF mutation frequencies in these cancer types were low (13 and 10, respectively).

## Association of BRAF variant class and mutant allele fraction

The mutant allele fraction (MAF) of genetic variants in bulk tumor sequencing data is a complex parameter reflecting tumor cell content, clonal architecture of the tumor and ploidy of the tumor genome. Oncogenic drivers generally show higher mutant allele fractions compared to passenger mutations. The author set out to investigate the mutant

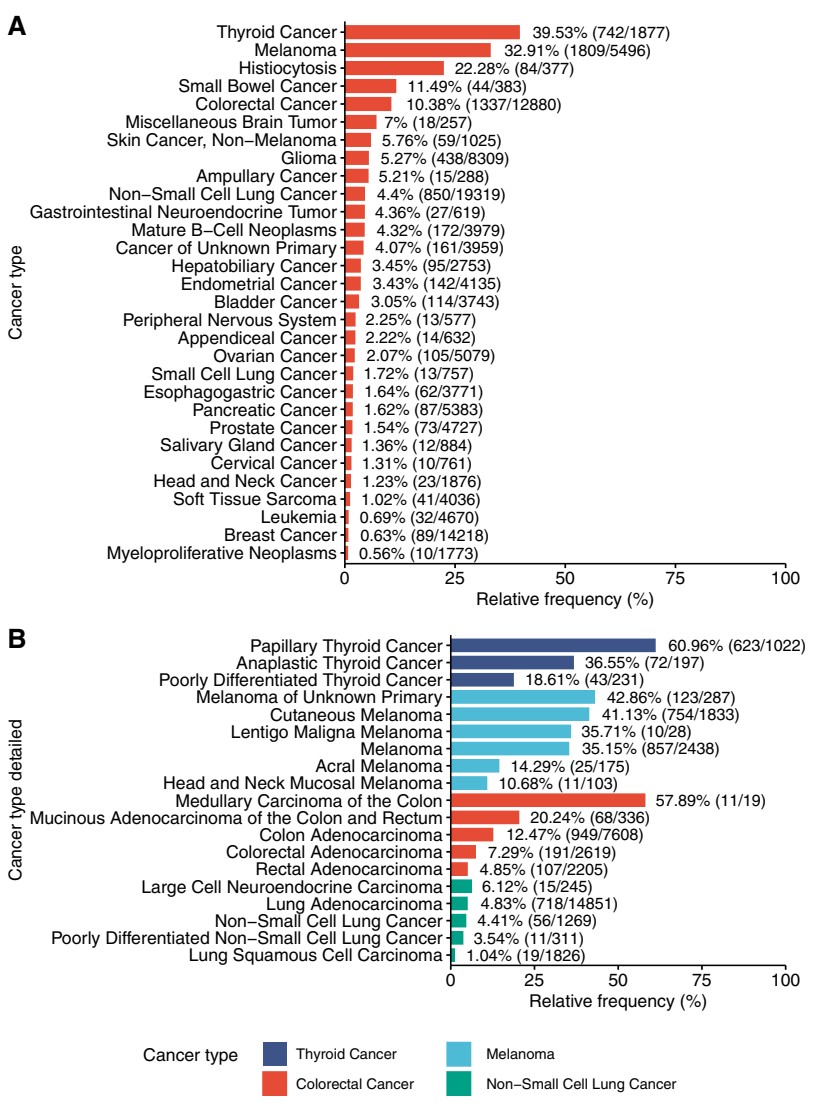

**Figure 2 Frequency of BRAF mutations in different cancer types and subtypes.** (A) Frequency of BRAF mutations in cancer types included in the Project GENIE dataset. (B) Frequency of BRAF mutations in subtypes of thyroid cancer, melanoma, colorectal cancer and non-small cell lung cancer.

allele fraction of BRAF variants assigned to class 1, 2 and 3 as well as variants that could be not assigned to a specific variant class based on literature mining. The data revealed that BRAF variants assigned to class 1 showed the highest median MAF across all samples in the dataset (Fig 4A). Class 2 variant displayed a significantly lower MAF than class 1 variants. Class 3 variants and reoccurring variants of unknown significance showed an even lower MAF.

These observations, however, might be partially attributed to the difference of mutant allele frequencies across cancer types. For example, in the case of non-small cell lung cancer, the median MAF was significantly lower for class 1 variants compared to class 2 variants (Fig. 4B).

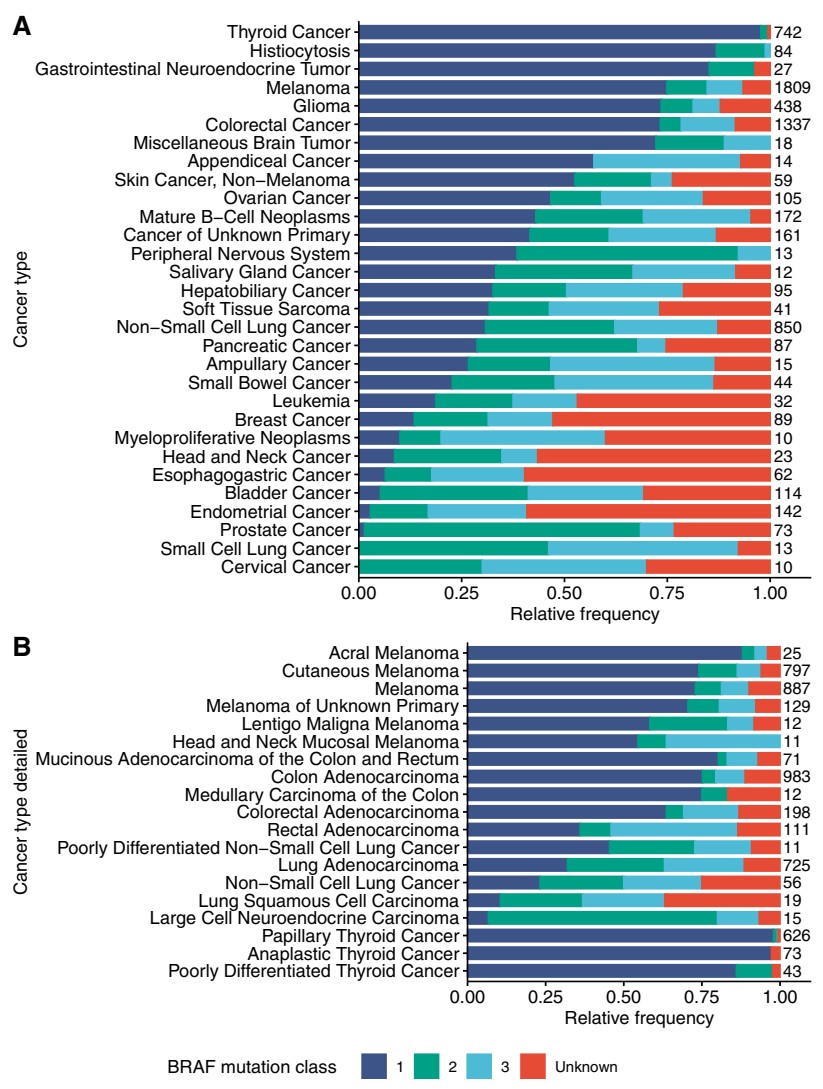

**Figure 3 Distribution of BRAF mutation classes across cancer types and subtypes.** (A) Distribution of BRAF mutation classes across cancer types included in the Project GENIE dataset. (B) Distribution of BRAF mutation classes in subtypes of thyroid cancer, melanoma, colorectal cancer and non-small cell lung cancer.

## Genetic interactions of BRAF mutations are cancer type-specific

To investigate genetic interactions of oncogenic BRAF, variants the author performed a co-mutation analysis. The dataset was filtered for the indicated cancer types and random permutation was performed to generate a null hypothesis. Based on the null hypothesis, the expected frequency and expected relative frequency of co-occurrence of mutations in BRAF with mutations in other genes was calculated. The employed approach can account for the heterogeneous cancer samples and sequencing approaches included in the dataset. To account for the biological differences between cancer types, the analysis was performed separately for non-small cell lung cancer, melanoma, colorectal cancer and thyroid cancer.

In the thyroid cancer samples BRAF mutations co-occurred less frequently than expected with mutations in NRAS, HRAS, RET, PTEN, NF1 and KRAS. In contrast,
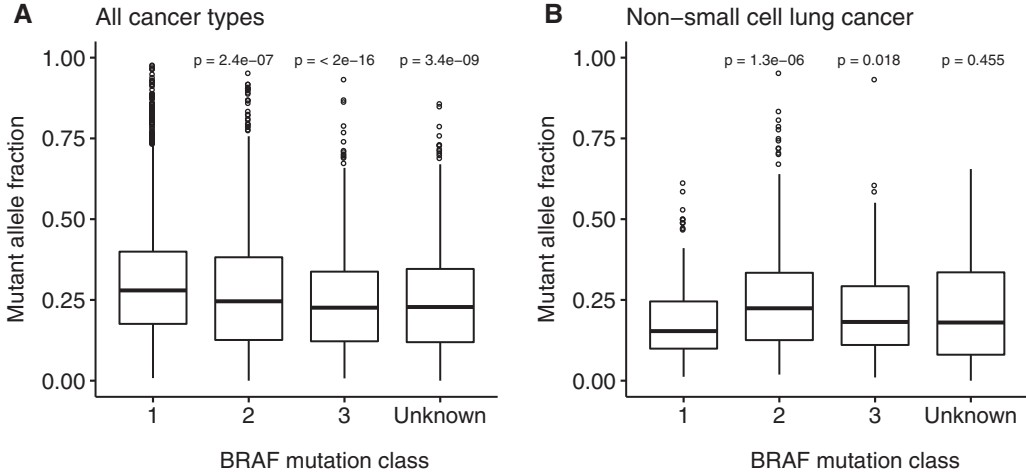

**Figure 4 Mutant allele fraction of BRAF mutations.** (A) Mutant allele fraction of BRAF mutations in all samples in the dataset. (B) Mutant allele fraction of BRAF mutations in non–small cell lung cancer samples.

mutations in BRAF and PIK3CA co-occurred more frequently than expected (Fig. 5A). In melanoma samples, NRAS mutations co-occurred much less frequently with BRAF mutations than expected. Also, mutations in GNA11, GNAQ, KIT, SF3B1, NF1 and TP53 were found less frequently in BRAF-mutated samples than expected. In contrast, PTEN was found slightly more frequently than expected in BRAF-mutated samples (Fig. 5B). In colorectal cancer samples, the author observed a trend towards mutual exclusivity for BRAF and KRAS mutations. Also, APC and TP53 mutations were found less frequently in BRAF-mutated samples than expected. In contrast, the author observed an enrichment of mutations in ARID1A, CREBBP, FAT1 KMT2A, KMT2D, NOTCH3, PTEN and RNF43 in BRAF-mutated samples (Fig. 5C). Non-small cell lung cancer samples with BRAF variants were less likely than expected to carry also alterations in EGFR and KRAS. The author also observed a similar trend for TP53. In contrast, variants in BRAF and ATR, FAT1, KEAP1, SETBP1, SETD2 and STK11 did more frequently co-occur than expected.

## The BRAF variant class defines genetic interactions

Next, the author wanted to investigate if the BRAF variant classes define the genetic interactions. To avoid interference by cancer type-specific mutations, only the non-small lung cancer samples in the GENIE dataset were used for these analyses. The null hypothesis was generated by randomly permutating variant class labels, and relative expected and observed co-mutation frequencies were plotted for each variant class (Figs. 6A–6C).

The conducted analyses revealed that the genetic interactions of mutant BRAF alleles are highly dependent on the variant class: BRAF class 1 mutations co-occured significantly more frequently than expected with AKT1 and SETD2 mutations. In contrast, BRAF class 1 mutations were found less frequently than expected in samples with EGFR, ERBB4, KEAP1, KRAS, STK11 and TP53 mutations (Fig. 6A). BRAF class 2 mutations co-occured more frequently than expected with ATR, EPHA3, NRAS, STK11 and WHSC1 mutations.

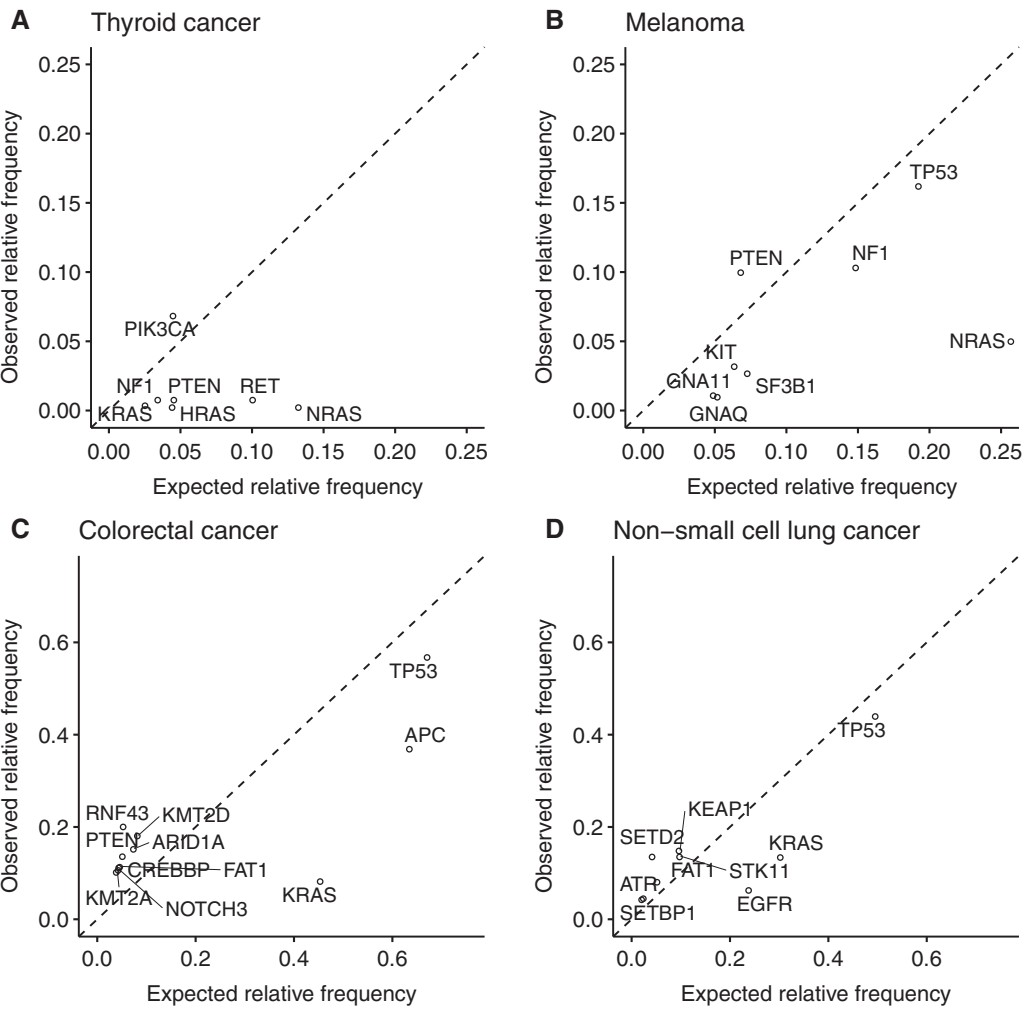

**Figure 5 Genetic interactions of BRAF mutations in thyroid cancer, melanoma, colorectal cancer and non-small cell lung cancer.** The scatter plots show the expected and observed relative co-occurrence frequency for mutations in the indicated genes with BRAF mutations in (A) thyroid cancer, (B) melanoma, (C) colorectal cancer and (D) non-small cell lung cancer.

In contrast, the author observed that EGFR and KRAS mutations are found less frequently than expected in samples with BRAF class 2 mutations (Fig. 6B). For, BRAF class 3 mutations, the author found a strong association with KEAP and STK11 mutations. As in the case of BRAF class 1 and 2 mutations, KRAS and EGFR were co-mutated less frequently than expected in samples with BRAF class 3 mutations (Fig. 6C). The author also created a heatmap displaying the ratio of observed relative frequency/expected relative frequency for all BRAF mutation classes to highlight the mutation class-specific genetic interactions (Fig. 6D).

# DISCUSSION

BRAF mutations are frequently found in cancer and considered oncogenic drivers. In this study, the author analyzed publically available data from the AACR Project GENIE to

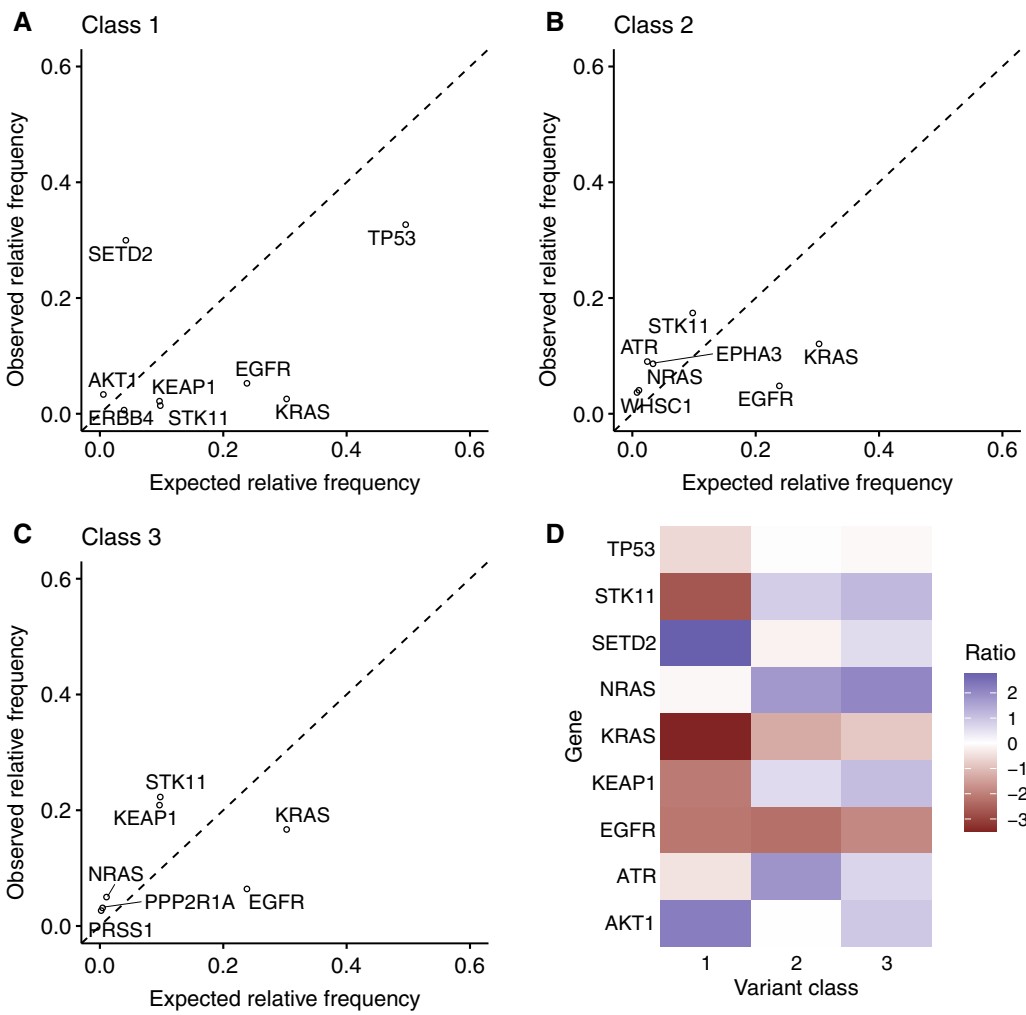

**Figure 6 BRAF mutation class-specific genetic interactions in non-small-cell lung cancer.** The scatter plots show the expected and observed relative co-occurrence frequency for mutations in the indicated genes with (A) BRAF class 1 mutations, (B) BRAF class 2 mutations and (C) BRAF class 3 mutations. (D) The heatmap displays the ratio of observed relative co-occurrence frequency and expected relative co-occurrence frequency for the indicated genes and BRAF mutation classes.

further understand clinical associations and genetic interactions of oncogenic BRAF alleles.

Initially, the author defined a subset of BRAF mutations that were found in ≥5 samples in the AACR Project GENIE dataset. 53.76% of these reoccurring mutations were previously reported in the literature and could be assigned to a functional mutation class. However, a large fraction of the identified BRAF mutations (92.01%) remained variants of unknown significance, highlighting the urgent need for further functional investigations. Notably, these variants of unknown significance accounted only for a small fraction of the samples (21%).

Next, the frequencies of BRAF mutations across cancer types were determined. The frequencies were similar to previously reported frequencies from other cohorts, with
minor deviations (*Holderfield et al., 2014*). However, due to the large number of samples and the diverse cancer types included in the AACR Project GENIE dataset, the author was also able to report BRAF mutation frequencies for rarer cancer types and subtypes.

The author also showed that the ratio of class 1, 2 and 3 variants varies across cancer types and subtypes and might reflect the oncogenic signaling in these cancer types. In prostate cancer, BRAF mutations occur at a relatively low frequency (1.54%) and previous studies have suggested that BRAF mutations in prostate cancer might be targetable (*Santos et al., 2020*). Notably, a majority of the BRAF mutations identified in prostate cancer could be assigned to class 2. Unfortunately, the clinical annotations in the project GENIE dataset were not sufficient to further clinically define this interesting subgroup of prostate cancer cases.

In cancer types with low frequencies of BRAF mutations such as leukemia, breast cancer, myeloproliferative neoplasms, head and neck cancer and esophagogastric cancer the author found high proportions of BRAF variants of unknown significance. The author speculates that these variants of unknown significance in these cancer types are passenger mutations and that the frequency of oncogenic and potentially targetable BRAF alterations in this cancer types is very low.

The author also attempted to correlate BRAF mutations class with mutant allele fraction, however the results from these analyses might be biased and were not consistent across cancer types.

Relying on the AACR Project GENIE dataset, the author was able to systematically investigate genetic interactions of oncogenic BRAF alleles in multiple cancer types. The analysis identified several interesting interactions, some of which have not been previously mentioned in literature. For all investigated cancer types, the author observed that mutations in upstream components of the MAPK/ERK signaling pathway, namely EGFR and RAS family proteins, co-occur much less frequently than expected with oncogenic mutations in BRAF. These finding had been previously reported in multiple studies (*Sensi et al., 2006*; *Li et al., 2014*).

For thyroid cancer the author found an interaction between BRAF and PIK3CA mutations which has been previously reported (*Charles et al., 2014*). For colorectal cancer, the author overserved co-occurrence of oncogenic BRAF mutations with mutations in the ubiquitin ligase RNF43. Recent papers have implicated oncogenic mutations in BRAF and RNF43 in the serrated neoplasia pathway of right-sided colorectal cancer (*Eto et al., 2018*; *Matsumoto et al., 2020*). The author also observed that in colorectal cancer samples, KMT2D mutations co-occur more frequently than expected with BRAF mutations. The author could not find a report that describes this co-occurrence in clinical samples however a recent mouse study demonstrated that KMT2D functions as tumor suppressor in BRAF V600E mutant melanomas (*Maitituoheti et al., 2020*).

Finally, the author investigated if the genetic interactions of BRAF alleles depend on the mutation class. For this analysis, only non-small cell lung cancer samples were used because this subgroup showed a balanced distribution of BRAF class 1, 2 and 3 mutations. This analysis confirmed that the BRAF mutation classes exhibit different genetic interactions that might reflect their signaling mechanisms. Interestingly, the author found,

that in NSCLC BRAF class 1 mutations co-occur with alterations in SETD2. This interaction had been already described before in a study focusing on non-small cell lung cancer (*Sheikine et al., 2018*). The author also observed an interaction of BRAF class 2 and 3 mutations with ATR, hinting to a possible connection of oncogenic BRAF and DNA damage response and replication stress signaling.

Taken together, the author employed publically available data from the AACR Project GENIE to explore the complex role of BRAF in cancer. The author believes that the presented analyses and data will be a valuable resource for researchers and clinicians focusing on BRAF biology and precision oncology.

## CONCLUSIONS

BRAF mutations are frequently found in cancer and considered oncogenic drivers. In this study, the author used data from the AACR Project GENIE to investigate clinical associations and genetic interactions of oncogenic BRAF alleles. The analyses demonstrate that the frequency of BRAF mutations greatly varies across cancer types and subtypes. Also, the distribution of BRAF mutations classes is highly unequal across cancer types and subtypes. Because of the large number of samples in the AACR Project GENIE dataset, the author was also able to systematically assess mutational co-occurrence of BRAF mutations with mutations in other genes in multiple cancer types. These analyses allowed to identify several interesting molecular subgroups (*e.g.*, colorectal cancer with BRAF and RNF43 mutations), however, the limited clinical annotations in the dataset did not allow to further clinically define these subgroups. The author believes that the presented analyses and data will be a valuable resource for researchers and clinicians focusing on BRAF biology and precision oncology.

## ACKNOWLEDGEMENTS

The author would like to acknowledge the American Association for Cancer Research and its financial and material support in the development of the AACR Project GENIE registry, as well as members of the consortium for their commitment to data sharing. Interpretations are the responsibility of the study author.

### Funding

The author received no funding for this work.

### Competing Interests

The author declares that they have no competing interests.

### Author Contributions

- Sebastian A. Wagner conceived and designed the experiments, performed the experiments, analyzed the data, prepared figures and/or tables, authored or reviewed drafts of the article, and approved the final draft.

## Data Availability

All source code and data tables are available in the Supplemental Files.

## Supplemental Information

Supplemental information for this article can be found online at http://dx.doi.org/10.7717/peerj.14126#supplemental-information.

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
