# Peer review of "Clinical associations and genetic interactions of oncogenic BRAF alleles"

_PeerJ, doi:10.7717/peerj.14126_

## Round 0.1 · original submission · Minor Revisions

The reviewers provided some very good opinions. The authors should revise these comments carefully, to reinforce the conclusions of this paper.

Reviewer 1 ·

Basic reporting

This is a highly interesting and well-conducted study. The manuscript is easy to follow. I have only a few suggestions, mainly concerning literature references, that could be adressed to further improve the manuscript.

1. line 46. This is a great but a bit outdated reference for an introduction. I would rather recommend to cite one of the more contemporary reviews discussing RAS mediated RAF activation, e.g. Cook & Cook (2021), PMID: 33367512

2. In relation to BRAF fusions, Ross et al (2016), PMID: 26314551 could be cited in addition to Jones et al. as this article provides an excellent overview about the spectrum of BRAF fusions

3. line 98 -101. This sentence should be rephrased to avoid confusion with the previous literature. The author cites Kordes et al (2016) in the context of pan-RAF inhibitors and in relation to kinase-inactive class 3 BRAF mutants and the use of pan-RAF and CRAF inhibitors. This concept is correct and has demonstrated preclinically (Smalley et al (2009), PMID: 18794803) and has already been pursued as an individual treatment option in a melanoma with a bona fide kinase-inactivating D594G mutation (Hoefflin et al, 2018, PMID: 32913998). In the cited paper, however, Kordes et al (2016) primarily investigated BRAF F595L, which they and previously two other groups (cited therein) classify as kinase-active and responsive to pan-RAF inhibitors.
Maybe the sentence could be rephrased as follows: There is also preclinical evidence demonstrating activity of pan-RAF and CRAF inhibitors in tumors with class 2 and inactivating class 3 mutations (Kordes et al., 2016; Smalley et al., 2009, Hoefflin et al, 2018).

Experimental design

No comment

Validity of the findings

No comment

·

Basic reporting

no comment

Experimental design

The manuscript focuses on exploring the clinical associations and genetic interactions of BRAF mutations across cancer types. The study seems promising. Authors used publicly available data sets and provided extensive analysis on BRAF mutations.


Major comments:
1. Studying the genetic interactions of BRAF mutations in cancer subtypes will provide a brief knowledge of differences in the aggressiveness of cancer subtypes. Therefore, authors should include BRAF mutation genetic interaction studies across cancer subtypes in this study as explained in Figure 5 and 6.

Minor comments:
1. Authors report that class I mutations lead to dimerization of BRAF instead of monomerization as shown in the abstract, line 22. It was opposite to the report by Dankner et al., 2018.

2. What are the 460 mutation frequencies (class 2) as shown in Figure 1B.

Validity of the findings

no comment

Reviewer 3 ·

Basic reporting

The article is written well, no additional comments.

Experimental design

The author should state which statistical test has been used in Figure 4. In the figure legend the author should consider stating what groups do the p values compare (For example - p value Class 1 vs Class 2 < 0.001)

Validity of the findings

No additional comments

Additional comments

If patient survival data is available for this dataset, it will be interesting to know the correlations between the different classes of BRAF mutation and patient prognosis in all, colorectal, melanoma, thyroid cancer and non-small cell lung cancer.

---

## Round 0.2 · Minor Revisions

The very minor comments from the Reviewer 1 should be further addressed.

Reviewer 1 ·

Basic reporting

All suggestions have been succesfully addressed and hence I recommend "accept". I have only one last minor suggestion:

I would rephrase the sentence "BRAF molecules with class 1 mutations signal as monomers and independent of upstream RAS." in lines 61/62 in "BRAF molecules with class 1 mutations can signal as monomers and independent of upstream RAS.". The "can" is important, because several labs have independently shown that BRAF V600E actually dimerizes quite effectively and is even more efficient as MEK kinase in its dimeric state. See Yuan et al (PMID: 29930381) and references therein. Of course, seminal studies have shown that BRAF V600E is only vulnerable to type I1/2 inhibitors in its monomeric state and artificial monomerization by introducing mutations into BRAF V600E have shown that it is still active in the monmoeric state, but we should avoid the impression that this oncoprotein always exists as a monomer in canecr cells.

I don't need to see the manuscript again, but I am available for further discussion, if necessary.

Experimental design

see my original comments

Validity of the findings

see my original comments

·

Basic reporting

No comments

Experimental design

No comments

Validity of the findings

No comments

Reviewer 3 ·

Basic reporting

No comments

Experimental design

No comments

Validity of the findings

No comments

---

## Round 0.3 · accepted · Accept

Thank you for your careful revision.